



# A study on the fragmentation of sulfuric acid and dimethylamine clusters inside an Atmospheric Pressure interface Time Of Flight Mass Spectrometer

Dina Alfaouri,[1] Monica Passananti,[1,2] Tommaso Zanca,[1] Lauri Ahonen,[1] Juha Kangasluoma,[1] Jakub Kubečka,[1] Nanna Myllys,[3] and Hanna Vehkamäki[1]

[1]Institute for Atmospheric and Earth System Research / Department of Physics, University of Helsinki, 00100 Finland
[2]Dipartimento di Chimica, Università di Torino, Torino, 10125, Italy
[3]Department of Chemistry, University of Jyväskylä, Jyväskylä, 40014, Finland

*Correspondence to*: Hanna Vehkamäki (hanna.vehkamaki@helsinki.fi)

**Abstract.** Sulfuric acid and dimethylamine vapours in the atmosphere can form molecular clusters, which participate in new particle formation events. In this work, we have produced, measured and identified clusters of sulfuric acid and dimethylamine using an electrospray ionizer coupled with a planar differential mobility analyser, connected to an atmospheric pressure interface time-of-flight mass spectrometer (ESI–DMA–APi-TOF MS). This set-up is suitable for evaluating the extent of fragmentation of the charged clusters inside the instrument. We evaluated the fragmentation of 11 negatively charged clusters both experimentally and using a statistical model based on quantum chemical data. The results allowed us to quantify the fragmentation of the studied clusters and to reconstruct the mass spectrum removing the artifacts due to the fragmentation.

**KEYWORDS:** APi-TOF MS, planar differential mobility analyser, electrospray ionization, statistical model, cluster fragmentation.

## 1 Introduction

Our climate is heavily impacted by atmospheric aerosol particles. These particles also play an important role in our daily lives. They determine the quality of the air we breath and thus affect our health directly (Hirsikko et al., 2011; Zhao et al., 2021). The majority of particles in the Earth's atmosphere are formed from gaseous precursors. Both laboratory and field measurements indicate that sulfuric acid, often with various amines acts as the main precursor for atmospheric new particle formation events by forming nanometer-scale clusters (Chen et al., 2012; Kürten et al., 2014; Mäkelä et al., 2001; Qiu and





Zhang, 2013; Smith et al., 2010; Thomas et al., 2016; Zhao et al., 2011). In recent years, developments in high resolution mass spectrometry have facilitated an increasing understanding of the chemical composition, concentration and stability of these molecular clusters. A central tool in detecting the elemental composition of these clusters is the Chemical Ionization Atmospheric Pressure interface Time-Of-Flight Mass Spectrometer (CI–APi-TOF MS) (Jokinen et al., 2012; Yao et al., 2018). However, due to the lower stability of clusters in comparison to molecules, clusters are more susceptible to

fragmentation and/or evaporation caused for example by ionization process, low pressure environments and high-energy collisions inside the instrument. Previous studies have shown that theoretical models often predict higher cluster concentrations compared to APi-TOF measurements (Kurtén et al., 2011; Olenius et al., 2013). Cluster fragmentation processes inside the instrument (Olenius et al., 2013) have been speculated to be an explanation for this difference.

   Our recent studies have made considerable progress in understanding the transformation of clusters inside the APi and in

simulating collision induced cluster fragmentation (CICF) (Passananti et al., 2019; Zanca et al., 2020; Zapadinsky et al., 2019). One of these studies (Passananti et al., 2019) investigated the fate of sulfuric acid trimer ions ($(H_2SO_4)_2HSO_4^-$) inside an APi-TOF MS by exploring the effects of the voltages applied in the APi chambers on the CICF, and identifying the regions of the APi in which the fragmentation is most likely to occur. Experimental results were found to be in good agreement with a theoretical model describing the CICF (Zapadinsky et al., 2019). This model simulates the motion of the

charged clusters and the energy exchange with the carrier gas molecules inside the APi-TOF MS based on statistical principles, combined with energy level data from quantum chemical calculations. The simulated dynamics are defined by the electric fields inside the chambers of the instrument and the random collisions of the charged clusters with carrier gas molecules (Zapadinsky et al., 2019).

   In this study, we extend our previous work to atmospherically relevant two-component clusters consisting of sulfuric acid

and dimethylamine. Due to their varying size and shape, different clusters tend to have different electrical mobilities. We use a planar-Differential Mobility Analyser (planar-DMA) (Amo-González and Pérez, 2018) to utilize this fact and select only one (known) cluster type at a time to enter the APi-TOF. We use an instrumental set-up (Fig. 1) consisting of an ElectroSpray Ionizer (ESI) and planar-Differential Mobility Analyser (planar-DMA) coupled with an Atmospheric Pressure interface Time-Of-Flight Mass Spectrometry (APi-TOF MS).

Our main goals are to use this set-up to identify the clusters that are fragmented inside the APi-TOF MS, and to quantify the fragmentation. We also compare our findings to theoretical fragmentation probabilities predicted by the CICF model (Zapadinsky et al., 2019). The combination of experimental and modelling data, allow us to reconstruct a mass defect plot of the detected cluster ions, removing the artifacts due to the fragmentation.






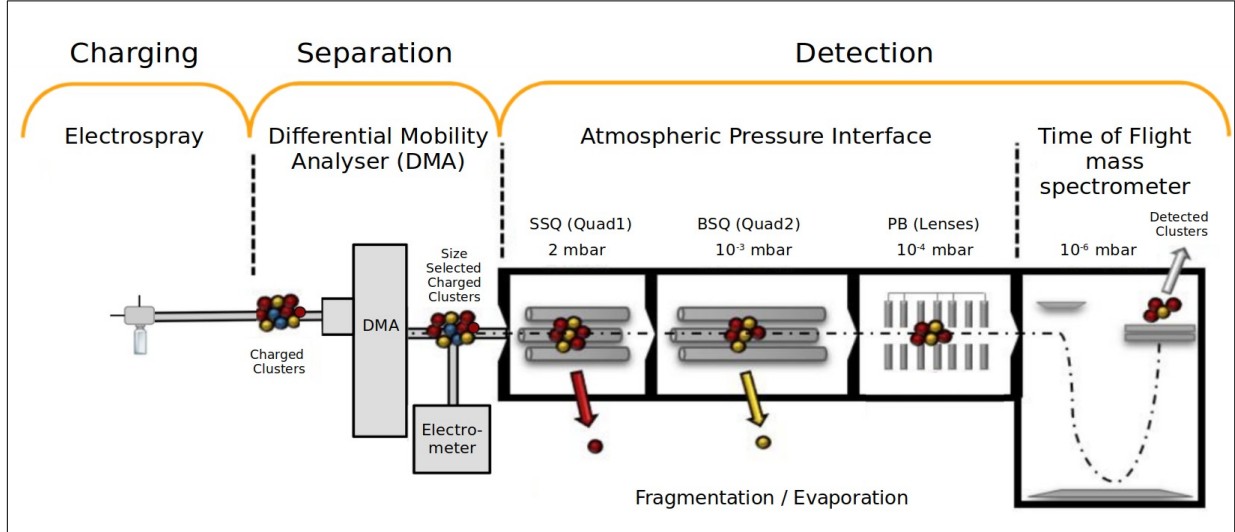

Figure 1. Schematic figure (not to scale) representing the experimental set-up of an electrospray ionizer and planar differential mobility analyser, connected to an atmospheric pressure interface time-of-flight mass spectrometer (ESI–DMA–APi-TOF MS). Figure modified from (Passananti et al., 2019).

## 2 Methodology

### 2.1 Experimental Set-up

The APi-TOF MS is an Atmospheric Pressure interface connected to a Time Of Flight mass spectrometer (Tofwerk). The TOF MS allows for the unambiguous identification of ions and clusters composition due to a resolving power up to 3000 Th/Th (Junninen et al., 2010). The APi part acts as a guide for the charged clusters from ambient pressure into high vacuum inside the TOF (~$10^{-4}$ mbar). Through the APi-TOF, charged clusters are subjected to a series of applied voltages (TOF Power Supply (TPS) voltages) which guide and focus them. These voltages hugely impact the fragmentation of the charged clusters and the instruments transmission. The planar-DMA is in turn connected to an electrometer and finally to the APi-TOF MS.

Molecular ions are generated using an ESI from a solution of 100 mM/100 mM dimethylamine/sulfuric acid in methanol and water with a ratio of 4:1 v:v. The sample is negatively charged using an electrode inserted in the liquid solution and then the charged sample is electrosprayed into the planar-DMA P5 (SEADM) with a sheath flow of $N_2$ carrier gas to separate charged clusters (with diameters up to a few nanometers) according to their electrical mobility. Thus, if we apply a certain voltage at the planar-DMA, then only clusters of one electrical mobility will be passed onward. In the planar-DMA two types of scans are conducted; full voltage scan, and fixed voltage scans. Full voltage scans are done within a range of −900 to −2900 V with





a voltage step of 5 V. Fixed voltage scans are done at the voltages where dimethylamine and sulfuric acid clusters appeared. The planar-DMA is in turn connected to an electrometer and finally to the APi-TOF MS where the clusters are detected. Further details on the experimental procedure are found in the Supporting Information (SI).

**2.2 Cluster Fragmentation Simulation**

We simulated the fragmentation of sulfuric acid and dimethylamine clusters using our statistical model (Zapadinsky et al.,
2019). As mentioned above, this model describes the motion of the charged clusters through the APi-TOF MS and the energy exchange caused by collisions between  the charged clusters and the carrier gas molecules. These collisions may cause the fragmentation of the cluster ions inside the instrument if they convey sufficient amount of energy (Zapadinsky et al., 2019). The model needs as input data on the experimental conditions (temperature, and voltages and pressures inside the APi chambers), and information about the (vibrational and rotational) energy levels which are used to evaluate the densities
of states. These latter were obtained using quantum chemistry data from calculations carried out within our group (Myllys et al., 2019), where vibrational frequency analysis were carried out at the ωB97X-D/6-31++G(d,p) level of theory). Further details on the model and quantum chemistry calculations are given in the SI.

**3 Results & Discussion**

Figure 2 shows a 2D plot of the combined and synchronized of signals from the DMA and the APi-TOF MS, with the
planar-DMA voltage on the *x*-axis, the cluster mass/charge ratio on the *y*-axis, and the signal intensity on a colour scale. This plot gives a convenient overview of the cluster fragmentation in the (negatively charged) sulfuric acid–dimethylamine system. For a given DMA voltage, in an ideal situation only singly charged clusters with a unique elemental composition (and thus mass) enter the APi-TOF MS. In the absence of fragmentation, this should result in one narrow peak in the mass spectrum and thus only one line in the 2D plot. Any deviation from this means that there are either multi-charged clusters,
singly charged clusters with different masses having the same mobility, and/or cluster fragmentation. As seen in Fig. 2, in our experimental conditions, the groups of peaks present are concentrated largely along one linear line, which means that we mainly observe singly charged clusters. In case of multi-charged clusters in 2D plots, several groups of peaks are concentrated along different linear lines (one line for each charge state), as an example of multi-charged 2D plots see Fig. 1(b) in (Larriba et al., 2014). Moreover, considering our sample composition and the resolution of the planar-DMA, the
likelihood of detecting singly charged clusters with different masses having same mobilities are low. This leaves us with cluster fragmentation, which can be highlighted from the 2D plots.

If a cluster entering the APi-TOF fragment, multiple signals are seen at the same voltage but at different mass/charge ratios. In Fig. 2, an example of a cluster and its fragment are shown (circled by dashed red lines).

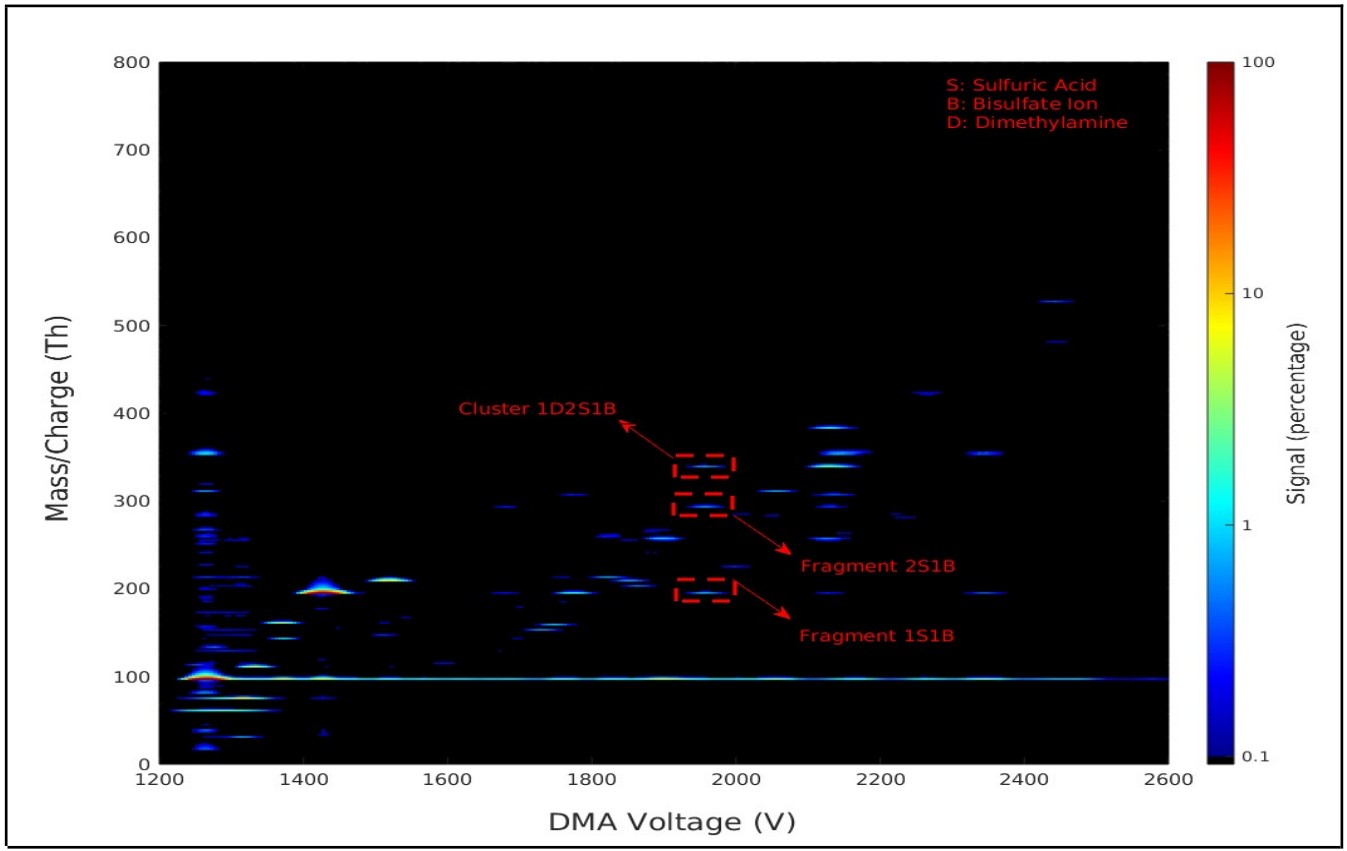

Figure 2. The 2D plot of the differential mobility spectrum and the mass spectrum of negatively charged sulfuric acid and
dimethylamine clusters generated by ESI. The plot show the mass/charge versus the DMA voltage with the signal intensity
as colour scale. Dashed red lines highlight the fragmentation of cluster 1D2S1B and its fragment 2S1B and 1S1B as an
example, where D = dimethylamine, S = sulfuric acid, and B = bisulfate ion.

Using the full voltage scan mode, it is possible to detect all negatively charged clusters of dimethylamine and sulfuric acid
produced by the ESI, within the scanned DMA voltage range given the APi-TOF transmission is good enough. Figure 3
shows a mass defect plot of all the 11 dimethylamine and sulfuric acid charged clusters produced and detected in our system.
For simplicity, throughout the whole paper we refer to sulfuric acid as S, dimethylamine as D, bisulfate ion as B and clusters
as for example 2D2S1B, which corresponds to a cluster of two dimethylamine molecules, two sulfuric acid molecules and
one bisulfate ion. The majority of the charged clusters had either a 1:1 ratio of sulfuric acid and dimethylamine with a bi-
sulfate ion attached, or a N+1:N ratio, i.e. with one more sulfuric acid than dimethylamine molecule (in addition to the bi-
sulfate ion). The smallest detected 1:1 ratio cluster is 2D2S1b, we do not observe 1D1S1B cluster probably due to its lower
stability compared to a larger cluster. This is in agreement with the computed trend of stability of negatively charged sulfuric
acid–dimethylamine clusters (Myllys et al., 2019). Moreover, our detected clusters are similar to those detected in a previous





study of the same sulfuric acid and dimethylamine solution (Thomas et al., 2016).   produced in gas-phase chambers
experiments (Almeida et al., 2013; Kürten et al., 2014) and  in ambient measurements (Yao et al., 2018).

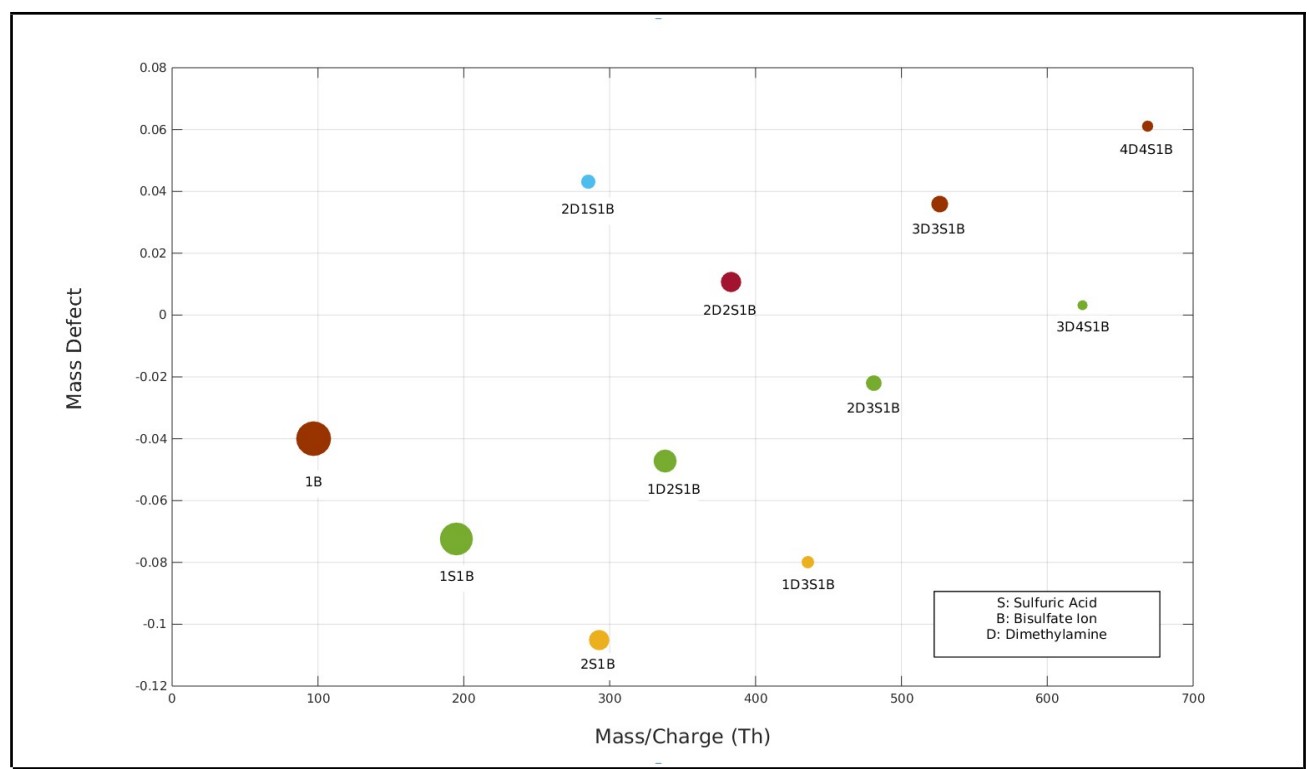

Figure 3. The mass defect plot of sulfuric acid and dimethylamine clusters detected by the APi-TOF MS. The circle size
reflects the intensity of the detected clusters. Clusters of the same colour have the same sulfuric acid:dimethylamine ratio.

The full voltage scan and the 2D plot (Fig. 2) are useful for providing a fast qualitative interpretation of the data. However,
for an in-depth analysis of the data and a quantitative measure of the fragmentation or survival probability of each cluster
type, experiments with fixed voltage and longer data acquisition times are needed. In a fixed voltage scan experiment a
single cluster type is selected, and the mass spectrum is recorded for that specific cluster. We performed fixed voltage scan
experiments for each cluster type to identify the fragmentation pathways and quantify the survival probability. For the
clusters not fragmented inside the APi, only the signal of the original cluster is observed in the mass spectrum. In case of
fragmentation, two or more signals are observed in the mass spectrum. For the larger clusters, we observed several
fragmentation pathways. Figure 4 shows the MS spectrum of the fixed scan experiment for the cluster 1D2S1B as an
example. In the MS spectrum there is the signal of the original cluster (1D2S1B) at 337.95 Th and there are three signals at
lower m/z representing fragmented cluster. Each signal is a cluster deriving from a different fragmentation pathway,
1D2S1B can fragment via these three pathways:



$$1D2S1B \rightarrow 2S1B + 1D \tag{R1}$$

$$1D2S1B \rightarrow 1S1B + 1S1D \tag{R2}$$

$$1D2S1B \rightarrow 1B + 1D2S \tag{R3}$$

We calculated the overall survival probability of 1D2S1B (using the ratio between the signal intensity of the parent cluster

and the sum of the parent and the fragmented clusters; all signal intensities have been corrected by the mass-dependent transmission of the APi-TOF) and the probability of fragmentation for each pathway. We also take into account an average background signal of the 1B ion (seen as a horzontal line in Fig. 2).

The fragmentation region inside the APi is relatively short (Passananti et al., 2019) and the daughter clusters is likely to leave this region before having a chance to fragment again, and thus we ignore subsequent fragmentation events. The

fragmentation pathways and the survival probability for each cluster are reported in the SI.

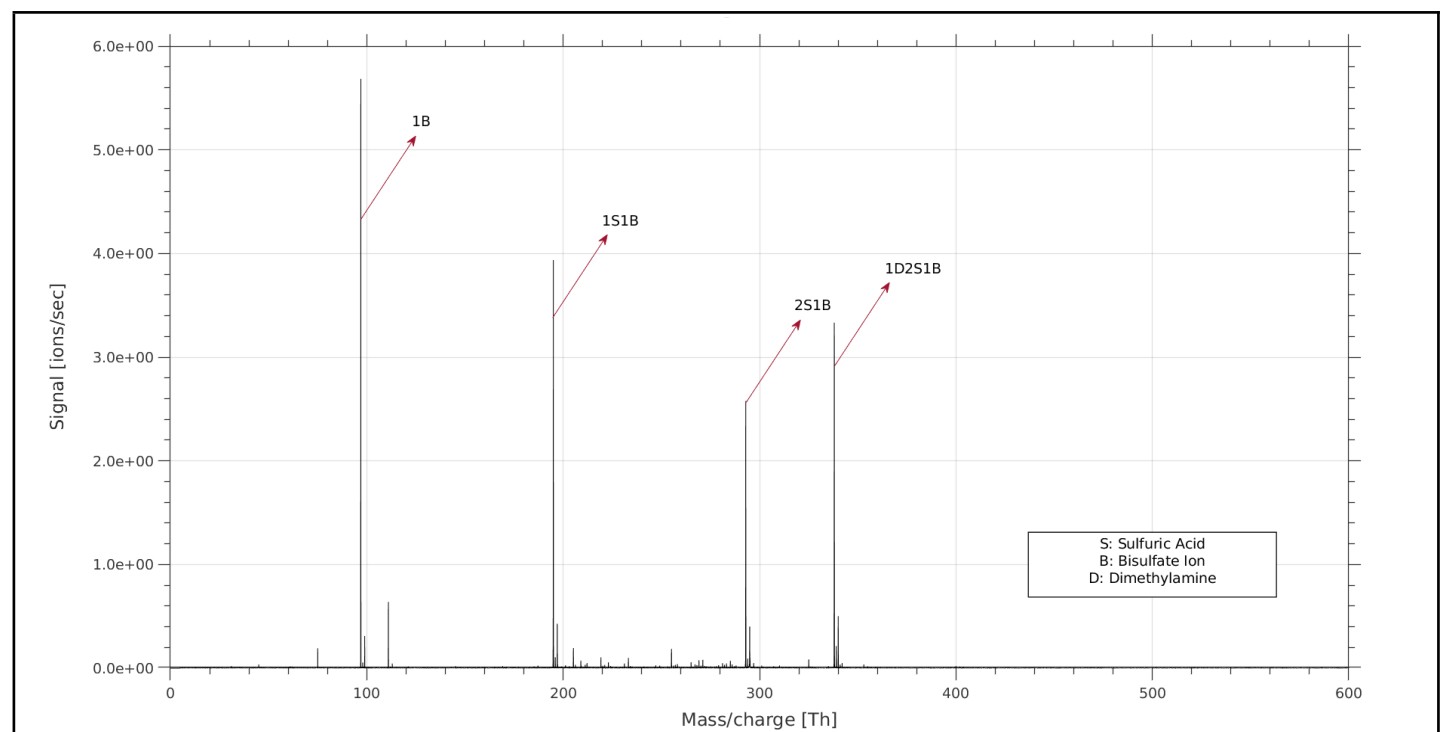

Figure 4. The MS spectrum of the fixed scan experiment for cluster 1D2S1B.

We compared experimental data with the survival probabilities calculated with the CICF model to understand the

fragmentation processes. To simulate the CICF inside an APi-TOF we needed to define all fragmentation pathways for each studied cluster. However, only single fragmentation pathways can be considered for each specific simulation. To identify the most probable fragmentation pathways we computed the reaction (kinetic) rate constants at different internal energies of the




cluster for each possible fragmentation pathway, and selected the pathways with the highest rate constants at the typical/average internal energy (see SI for more detailed information). All clusters except 1S1B, may fragment through at

least two different pathways and the number of pathways increases with the cluster size. Finally we calculated an overall survival probability for each cluster using the selected most probable fragmentation pathways.

Figure 5 shows the comparison of the overall survival probability according to experiments and model simulations for all studied clusters. For most of the clusters detected, the experimental and model results of the survival probability are in good agreement. There can be several reasons for the discrepancies in the survival probability between the experiments and the

model:

1) For some parent clusters multiple fragmentation pathways can occur simultaneously within the same experiment.

2) The fragmentation of a multi-charged cluster having the same mobility as a different singly charged cluster can produce the same fragments which leads to an underestimation of the experimental survival probability of the

studied singly charged cluster.

3) Clusters with very close mobilities can have overlapping signals which are difficult to separate.

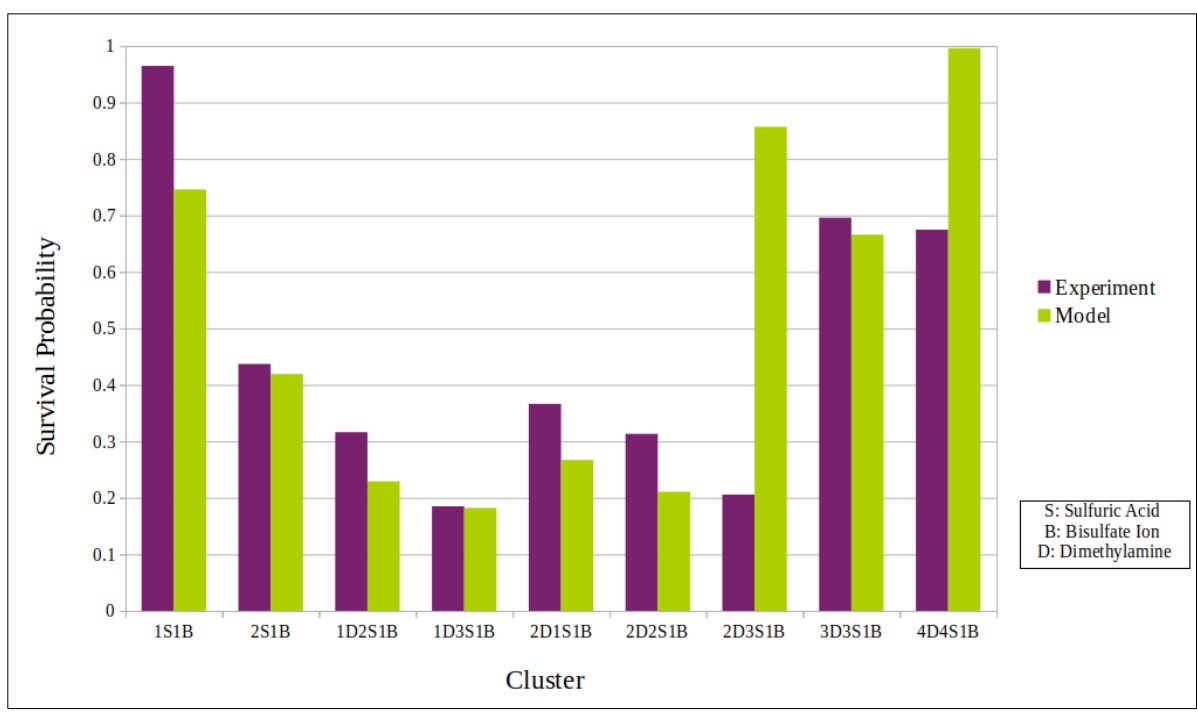

Figure 5. The overall survival probability of all sulfuric acid–dimethylamine clusters detected and calculated experimentally versus that simulated and calculated by our statistical model.



For most of the clusters the model underestimates the survival probability compared to the experimental results, which could be explained by reason 1 and/or 2 in the list above. Only for clusters 2D3S1B and 4D4S1B the model overestimates the survival probability and for these clusters there is a large discrepancy between the model and the experiments. The reason for this discrepancy might be the harmonic potential description of the vibrations used in deriving the energy levels of the cluster from quantum chemistry. For large clusters, ignoring the anharmonicity may result in overestimates for the survival probability.

Knowing the instrumental transmission (see SI section 6), cluster fragmentation pathways and survival probabilities allows for reconstruction of the mass defect plot removing the effects of fragmentation (Fig. 6). In particular, the intensity of a cluster was increased in case it has a survival probability lower than 1 and/or decreased if it was produced as a results of a fragmentation of another cluster. More details on the procedure to reconstruct the mass defect plot are reported in the SI. This procedure enables the removal of artifacts due to the fragmentation of clusters and gives more accurate information about the actual concentration and composition of detected clusters.

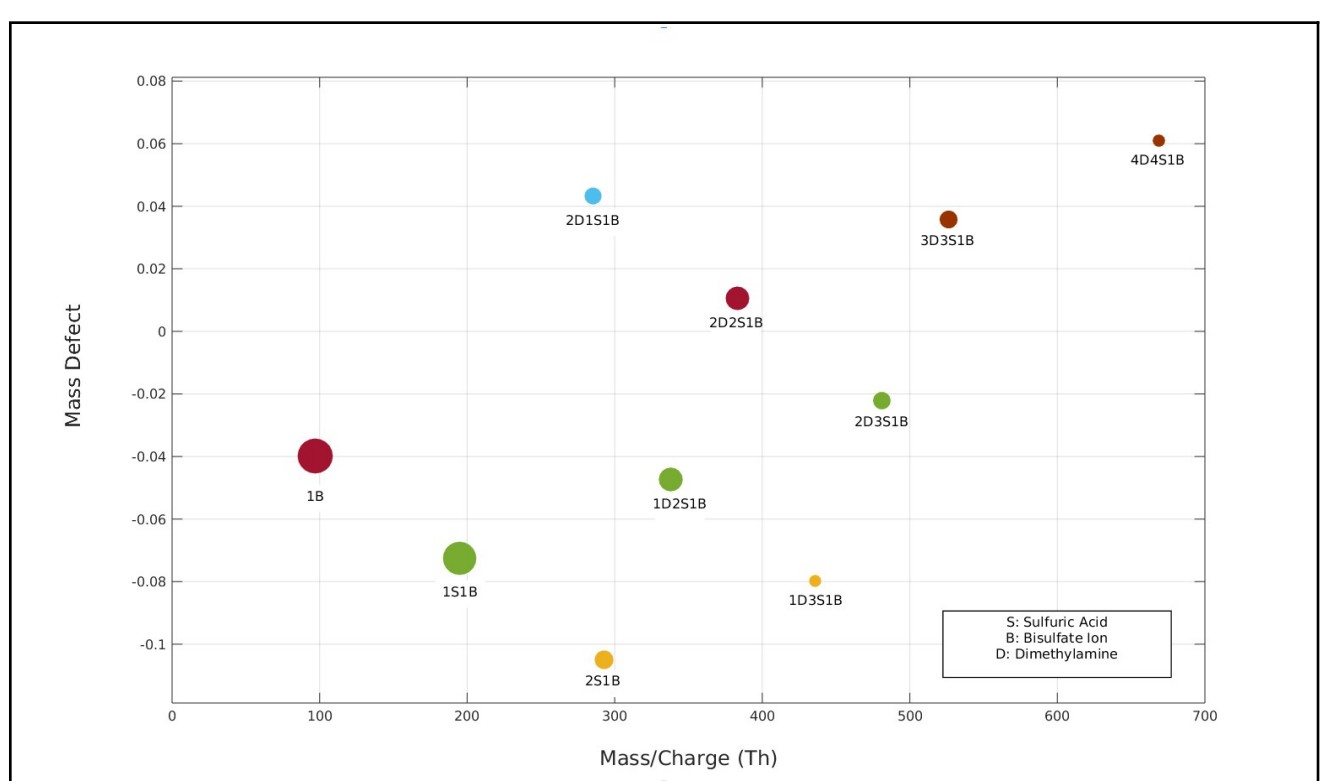

Figure 6. The reconstructed mass defect versus mass/charge (Th) plot based on that shown in Fig. 3 after accounting for all fragmentation processes. Note that cluster 3D4S1B is not seen here as it is only a fragmented product of cluster 4D4S1B.


## 4 Conclusion

In this work we tested our experimental set-up (ESI–DMA–APi-TOF MS), which consists of two high resolution instruments, and a first principle-based CICF model to study the fragmentation of atmospheric relevant clusters. We generated and identified 11 charged sulfuric acid and dimethylamine clusters, and for each of these clusters, we quantified the extent of the fragmentation inside the instrument both experimentally and using a statistical model. The results showed a good agreement between the experiment and the model, shedding light to the nature of the fragmentation processes within this instrument. Our study revealed that larger clusters may undergo multiple fragmentation pathways. Our data allowed us to reconstruct the mass spectrum (i.e. a mass defect plot) of the identified clusters so that we were able to define the original signal intensities of the detected clusters as if they had remained intact inside the instrument, removing artifacts due to the fragmentation. In the future, we anticipate that these proof-of-concept results can be extended also to other cluster-forming systems, and fragmentation corrections could be incorporated into standard data analysis tools related to these instruments. This kind of sophisticated data-analysis would significantly increase the accuracy of atmospheric cluster measurements allowing for a better understanding of the conditions that leads to new particle formation.

ASSOCIATED CONTENT

Supporting Information

ACKNOWLEDGMENTS

We thank the ERC Project 692891-DAMOCLES, Academy of Finland project 1325656, University of Helsinki Faculty of Science ATMATH Project, for funding, and the CSC-IT Center for Science in Espoo, Finland, for computational resources. We acknowledge Evgeni Zapadinsky for all his help, knowledge, discussions and his expert advice. We also acknowledge Theo Kurtén for his expert advice.

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
