# Peer review of "A study on the fragmentation of sulfuric acid and dimethylamine clusters inside an Atmospheric Pressure interface Time Of Flight Mass Spectrometer"

_Atmospheric Measurement Techniques, 2021_

## Author Response (AR1)

**RC1:** This work focus on sulfuric acid and dimethylamine clusters fragmentation.

Topic is interesting, allowing to discern the discrepancy between models and measurements of cluster concentration quantification.However, the topic is too far away from my knowledges and I can not evaluate it in a proper way.There is only one suggestion.

The experimental set-up description could be esaier to follow if the description followed the same order (from left to right) of description of the individual parts in the Figure 1.

**Reply:** Thank you very much for your comment. We will change the description of the individual parts to follow the same order from the schematic figure.
Please see below the modified text:

from line 66  '**2.1 Experimental Set-up**

As mentioned above, the ESI is coupled with a DMA which is in turn connected to an electrometer and finally to the APi-TOF MS. The APi-TOF MS is an Atmospheric Pressure interface connected to a Time Of Flight mass spectrometer (Tofwerk). The APi part acts as a guide for the ions and charged clusters from ambient pressure into high vacuum inside the TOF (~$10^{-4}$ mbar). The TOF MS allows for the unambiguous identification of ions and clusters composition due to a resolving power up to 3000 Th/Th (Junninen et al., 2010). Through the APi-TOF, charged clusters are subjected to a series of applied voltages (TOF Power Supply (TPS) voltages) which guide and focus them. These voltages hugely impact the fragmentation of the charged clusters and the instruments transmission. '

**RC2:** Alfaouri et al. conducted an investigation of the fragmentation of 11 sulfuric acid and dimethylamine clusters in the atmospheric pressure interface (APi) mass spectrometer. A DMA was used as the ion cluster classifier (prior knowledge of the ion). The results successfully reconstructed the mass spectrum by considering the fragmentation of clusters in the APi. The study presented a complete calibration procedure of ion cluster fragmentation and will be of great interest to the community. I recommend acceptance of the manuscript after some minor revisions:

**1.** Where is the background signal of the 1B ion from? Does it suggest that M1B->M+1B is a common fragmentation pathway in APi (M is an arbitrary cluster)? This should be clarified in the caption of Figure 2 or the main text.

**2.** How is the fragmentation rate threshold ($10^3$ s$^{-1}$) determined? Why is it the same for all the ion clusters? If the fragmentation time in the APi is the limiting step, how is that value estimated? I think this should be included in the main text.

**3.** This study presented the calibrated mass spectrum from DMA+APi-MS. But when we just have the APi-MS spectrum, how can this study be used to retrieve the concentrations (or relative intensities) of the clusters in the air sample that have the potential to fragment in the Api?

**Reply:** Thank you very much for your comments and questions. Below are the answers for each question individually.

1. We believe that since 1B is a highly stable ion, neutral evaporation or fragmentation in M1B$\rightarrow$ M+1B (M is an arbitrary cluster) is the reason for such horizontal lines appearing in the 2D plot. In general, we can observe this phenomena when evaporation or fragmentation leads to formation of highly stable ions or clusters. This will be included in the explanation of the figure.

We will change the text as reported below.

From line 102 'If a cluster entering the APi-TOF fragments, multiple signals are seen at the same voltage but at different mass/charge ratios. In Fig. 2, an example of a cluster and its fragment are shown (circled by dashed red lines). Moreover, many clusters upon entering the APi-TOF MS undergo neutral evaporation or fragmentation, especially when the produced cluster or ion is a highly stable one. This could result in a continuous horizontal line as seen in Fig.2 for the 1B ion where M1B$\rightarrow$ M+1B (M is an arbitrary cluster).'

2. The probability of fragmentation before the next collision is given at the end by two parameters: the collision rate and the fragmentation rate. Precisely, the probability to fragment before the next collision occurs is given by k/(k+c), where fragmentation rate is "k" and collision rate is "c". If k<<c, then we can exclude that a fragmentation would occur before the collision, because the probability is practically zero. The collision rate is approximately $10^7$ 1/s in the first chamber and $10^5$ 1/s in the second chamber, and we set the threshold for the fragmentation rate at 1000 1/s (which is about three orders of magnitude lower than the typical collision rate) because using this value we can safely assume that the cluster will not fragment before the next collision (the probability of fragmentation is ~ 0.1%). The threshold value has been chosen arbitrarily, considering 0.1% a reasonable probability limit for neglecting fragmentation. Of course, there's no precise value for the threshold, one could decide to set it at a lower value to have an even smaller probability of fragmentation.

We will add the following sentence in the supporting information.

SI from line 47 '… internal energy of the clusters. We set the threshold for the fragmentation rate at $10^3$ s$^{-1}$ (which is about three orders of magnitude lower than the typical collision rate) because using this value we can safely assume that the cluster will not fragment before the next collision (the probability of fragmentation is ~ 0.1%). Table S1 lists all….'

3. At the moment, it is not possible to measure the degree of fragmentation in a setup of APi-TOF MS only without the use of a differential mobility analyzer and an electrometer at the inlet of the system. It is, however, possible to calculate the degree of fragmentation in the APi-TOF using our statistical model. The degree of fragmentation depends on a lot of variables, mainly including the type of clusters being measured, the pressure inside the APi-TOF MS, the voltages across the different APi chambers, and the radio frequencies of the quadrupoles, etc.. We can assume that the degree of fragmentation for a specific cluster will be the same in an APi-TOF with the same voltage configuration and the same pressures (in the APi chambers) as used in our experiments. In case the APi-TOF is run with different voltage configuration and pressures it is possible to estimate the cluster fragmentation using our statistical model. Indeed, the input data of our model are the pressures in the APi chambers, the voltage configuration and the quantum chemistry data (energy levels and vibrational frequencies) of the studied cluster. We can also use inverse  mathematics and/or machine learning to back calculate the original cluster distribution from the detected one once the model has been validated against experiments.

In general, the lack of comprehensive studies on the quantification of fragmentation of different ambient clusters being measured is a major setback for retrieving realistic concentrations of clusters in air samples. It would be good to collect a large set of fragmentation data to be used as reference for common atmospheric clusters.

**RC3:** Review to "A study on the fragmentation of sulfuric acid and dimethylamine clusters inside an Atmospheric Pressure interface Time Of Flight Mass Spectrometer " by Alfaouri et al., AMTD 2021

This manuscript describes the study of collision induced cluster fragmentation inside an API-ToF. The authors chose negatively charged clusters of dimethylamine and sulfuric acid produced in an electrospray ion source and selected clusters using a differential mobility analyzer. The results are used identify cluster fragmentation degrees and pathways and to correct the abundance of ion clusters measured by the API-ToF. The paper is short and concise. It fits into the scope of AMT and I recommend publication after my comments listed below have been addressed.

**General comments:**

**Section 2.1:**
What makes the planar-DMA special here? I assume the selection for mobility diameter would work with any DMA model? Thus, I suggest to introduce the planar DMA only at the beginning of the experimental section and after that use simply "DMA", as you do later.

**Section 3:**
**How do I relate the information from Figure 2 to the selection of clusters in Figure 3?** Here an important piece of information is missing. A table showing the DMA voltages and the selected cluster(s) belonging to that voltage might help. Are the clusters shown in Figure 3 all clusters that are found in the experiment? If so, it follows that the same amount of clusters is displayed in Figure 2. But it appears as if there a a lot more in Figure 2. This deserves a more detailed explanation. See also the technical comment to Fig. 2 below.
**Lines 178-191:** If cluster size plays a role, then why is the discrepancy highest for the 2D3S1B cluster? For 3D3S1B you find good agreement.

**Technical comments:**

**line 53:** acronym planar-DMA has already been introduced two lines above
**line 73-74:** this sentence "The planar-DMA is in turn connected to an electrometer and finally to the APi-TOF MS" is repeated later in line 81.
**line 76:** Is "SEADM" the manufacturer and P5 the model type of the DMA?
**Fig 2:** This is a nice overview on the experimental results, but in the present form it is hard to read, it looks like "raw data" and is not very informative. In the present form it can be moved to the supplement. What the reader would like to know is how you find the $H_2SO_4$-DMA clusters from Fig 2 that were selected for further analysis. What information does the DMA voltage give us? I think that the voltage can be converted to a mobility diameter in nm.

**Supplement:**

**Line 20 and 22:** please give also city and country for SEADM, as you did for Tofwerk in line 23
**Line 26:** "The first two..."
**Line 78:** "arise"
**Line 83:** what is a "Herrmann-DMA"?

**Reply:** Thank you very much for your comments and questions. Below are the answers for each question individually.

**General comments:**

**Section 2.1:**
The ultrahigh resolution of the planar-DMA is the reason why we choose this device for studying the mobility of the clusters produced in our system, and complex cluster systems in general (especially for ambient measurements). The ultrahigh resolution of the planar-DMA allows for the

separation of ions and clusters with electrical mobilities close to each other, which would not be possible with some other DMAs. This paper (https://doi.org/10.1021/acs.analchem.8b00579), provides a good comparison of resolving power between different types of DMA in Table 1. We will follow your suggestion and introduce the planar DMA only at the beginning of the experimental section.

**Section 3:**
We explain how to relate information from Figure 2 to clusters in Figure 3 in the answer of Technical comments Figure 2.
We will add a column in the Table S2 (supporting information) including the DMA voltage for each identified cluster. Figure 3 shows all Sulfuric acid/Dimethylamine clusters produced and detected in our system. The discrepancy between the number of clusters in Figure 2 and 3 comes from the fact that in any system like ours a lot of other clusters (mostly impurities) are produced. Those clusters are not relevant for this fragmentation study so they were not mentioned in the text. But an explanation is now provided in the description of figure 3.
From line 115 'Figure 3 shows a mass defect plot of all the 11 charged dimethylamine and sulfuric acid clusters produced and detected in our system. Other clusters or impurities are not shown in the figure as they are not relevant for this study. The DMA voltages and the m/z values for each detected clusters are reported in Table S3 of the Supporting Information.'

**Lines 178-191:** We are not sure about the reason for this larger discrepancy for the 2D3S1B cluster. It could be a combined reason of both the experimental and model uncertainty in the evaluation of the survival probability.
As reported in the manuscript there can be several reasons for the discrepancies in the survival probability between the experiments and the model:
1. For some parent clusters multiple fragmentation pathways can occur simultaneously within the same experiment.
2. The fragmentation of a multi-charged cluster having the same mobility as a different singly charged cluster can produce the same fragments which leads to an underestimation of the experimental survival probability of the studied singly charged cluster.
3. Clusters with mobilities very close to each other can have overlapping signals which are difficult to separate.

On one side the trend in the clusters 2D2S1B, 3D3S1B, and 4D4S1B could be explained by the increasing role of anharmonicity with cluster size, while 2D3S1B does not fit to this trend due to different ratio of dimethylamine to sulfuric acid molecules. 2D3S1B has less hydrogen bonds since there are only 2 dimethylamine molecules in the cluster, this corresponds to lower bond network which may lead to a higher uncertainty. In addition to that, as indicated by table S1 in the supporting information, for the cluster 2D3S1B more simultaneous fragmentation pathways were observed experimentally in comparison to all other clusters. This contributes to a higher uncertainty in the experimental survival probability calculation for this cluster.

**Technical comments:**

**line 53:** Done.
**Line 73-74:** Adjusted.
**line 76:** Yes, and it has been adjusted in the text.
**Fig 2:** The main advantage of figure 2 is the visualization of the clusters and their fragments over the range of DMA voltages scanned for this experiment. This way of visualizing the data allows us to evaluate the presence of multi-charged compounds, the presence of fragmented clusters and the range of m/z and mobility of the clusters produced in the ESI. The raw data are the mass spectra and the DMA spectra, which we combine (after synchronizing the instruments) through a Matlab script

to visualize the data and provide an overview of the results. To identify the clusters (in our case sulfuric acid-dimethylamine clusters) we analyze the MS data using TofTools (a Matlab GUI created in our institute) and for each of the identified cluster we retrieve the DMA mobility data through a Matlab script. Summarizing, the identification of the 11 clusters of sulfuric acid and dimethylamine in our system is done using the mass spectrum. Following that, the DMA voltage at which each cluster appeared was determined by evaluating the electrical mobility of each of those clusters.

Finally, to quantitatively study the fragmentation, experiments at fixed voltage scans of the DMA are performed. This allowed only clusters of that specific mobility to enter into the APi-TOF MS which allowed us to study the fragmentation and fragmentation pathways of the individual clusters (an example is reported in Figure 4).

We will add the following explanations in the main text to better explain the use of figure 2 and the connection with figure 3:

From line 95 '…colour scale. This type of data visualization allows to evaluate the presence of multi-charged compounds, the presence of fragmented clusters and the range of m/z and mobility of the clusters produced in the ESI.  Indeed, this plot…'

From line 115 '…good enough. To identify sulfuric acid-dimethylamine clusters the MS data have been analysed and clusters are reported in Figure 3. In particular, Figure 3 shows…'

**Supplement:**

**Line 20 and 22:** Done.
**Line 26:** Done.
**Line 78:** Done.
**Line 83:** It is a type of high resolution DMA that was developed at Yale-University. This article (https://doi.org/10.1080/02786826.2016.1142065) talks more about this specific type of DMA. We will add this reference in the text.